# Impact of Elevated Temperatures on Strength Properties and Microstructure of Calcium Sulfoaluminate Paste

**DOI:** 10.3390/ma14226751

**Published:** 2021-11-09

**Authors:** Konrad A. Sodol, Łukasz Kaczmarek, Jacek Szer, Sebastian Miszczak, Mariusz Stegliński

**Affiliations:** 1Faculty of Mechanical Enginnering, Institute of Materials Science and Engineering, Lodz University of Technology, 90-924 Łódź, Poland; lukasz.kaczmarek@p.lodz.pl (Ł.K.); sebastian.miszczak@p.lodz.pl (S.M.); mariusz.steglinski@p.lodz.pl (M.S.); 2Department of Building Physics and Building Materials, Faculty of Civil Engineering, Architecture and Environment Engineering, Lodz University of Technology, 90-924 Łódź, Poland; jacek.szer@p.lodz.pl

**Keywords:** calcium sulfoaluminate, high-temperature, microstructure, strength, green binder, cement

## Abstract

This article is motivated by civil fire safety. Fire-prevention engineering demands a wide range of information about building materials including alternative cements, for instance CSA-cement. Because of exposure of the cement-base material to a high temperature, its strength properties deteriorate due to dehydration connected with phase and microstructure changes. Previous research indicated that the main endothermic reaction of CSA-based composite, dehydration of ettringite, might be used as a cooling system for a metal structure during fire-load. This article examines visual assessment, microstructure, density, as well as flexural and compressive strength parameters of CSA-based composite after isothermal heating at temperatures from 23 °C to 800 °C. The results of SEM/EDS investigations showed that the calcium sulfoaluminate paste may start partially re-sintering above 600 °C. Mechanical tests revealed significant reduction of strength parameters but residual compressive strength was maintained in the whole temperature range e.g., 8 MPa at 800 °C. Additionally, visual assessment of the specimens indicated that it might be possible to predict the material temperature heating based on the specific surface color. These findings add to the evidence of general knowledge about CSA hydrates.

## 1. Introduction

One of the most popular binders in civil engineering is cement. Because of estimation that cement manufacturing release to the atmosphere 5–7% global CO_2_ per year [1] it is highly important to reduce carbon-dioxide emission in this branch of economy. The example of a binder with lower footprint than ordinary Portland cements (OPCs) is calcium sulfoaluminate (CSA) cement. This binder is generally considered as a green alternative to OPC [2]. Its CO_2_ emission is reduced because of the lower temperature needed for the sintering process and energy for klinker milling during production in comparison to OPC manufacturing [3]. Standard ASTM C-845-96 [4] describes expansive cements (CSA). The main hydrated calcium sulfoaluminate phases are ettringite, monosulfite and alumina hydroxide [5]. Previous research indicated that hydration of ettringite caused tensile stress in materials because of high expansiveness of this phase [6]. Because of ye’elemite and gypsum hydration, highly-compressive strength cement stone is created, where compressive strength may be higher than in OPC [7]. Strength and performance parameters are very important for special civil projects applications, therefore CSA cements should also be tested in extreme conditions, such as very low or high temperatures, to ensure safety in all situations.

In general, high temperature exposure causes a series of physio-chemical changes in cementitious materials. Previous issues show that strength parameters decrease, the spalling effect may destroy material, pores rise and material loses water during the dehydration of phases [8]. The influence of high temperature on cement composites was the subject of experimental and theoretical extensive studies [9,10,11,12,13,14]. Most of them considered Portland cement-based materials. Thermo-hydro-mechanical consideration with temperature influences on CSA-based material have not been fully analysed. The findings describing phase transition of CSA materials during high temperature are presented in Table 1.

Sodol et al. [17] showed that CSA-based, OPC-CSA-based composite had a higher endothermic sum of phase transition during high temperature than OPC and paste. They pointed out that better understanding of an endothermic reaction is important from a cooling mechanism point of view. During fire-temperature exposure, a cooling mechanism might protect metal structures as infills or covers. The experimental data indicate that the cooling mechanism of CSA-based material is more efficient than that of OPC during 30 min and 60 min. The previous research shows that Portland cement material is heat resistant up to 400 °C [18], while CSA is heat resistant up to 150 °C [19]. Due to reversible ettringite phase transition of CSA-based material, it is possible to use it as heat storage system [20].

Because of fire safety engineering considerations, it is necessary to expand the data base about high-temperature influence on alternative binders, for example CSA-based materials. This paper refers to CSA paste behavior during exposure to elevated temperature of 800 °C. It shows the strength parameters and density as a temperature function. The findings extend the current stage of knowledge about microstructure of CSA paste loaded by temperature and describes principal visual assessment rules of colors after high-temperature exposure.

## 2. Materials and Methods

### 2.1. Specimen Preparation

Specimens were prepared by mixing water from a water framework with CSA cement, (series AliCEM Green), where the main binder was a mixture of CSA cement clinker and gypsum in 4:1 ratio (chemical composition is showed in Table 2).

The cement/water ratio was 0.5. The temperature of water used in mixture was 9–10 °C. We created 24 specimens in 40 mm × 40 mm × 160 mm forms, according to the PN-85/B-04500 standard [21]. In the first step, ingredients were combined and mixed by using electrical stirrer. Then, liquid cement paste was poured to tripartite forms which were subjected to vibration. After removing excess paste, the specimens were secured by wet gauze and filter paper to reduce the water evaporation and left for 24 h. Following this, specimens were demolded and treated in a 100% humidity environment (wet conditioning) for 6 days. Then, the samples were treated in 65 ± 5% humidity environment for 21 days (air-wet conditioning). Finally, 28-day samples were heat-treated in an electrical atmosphere oven (NEOTherm, Zakład Elektromechaniczny, Wrocław, Polska). The heat-treatment regime, including: heating rate. isothermal heating, cooling rate, number of specimens and their shape was maintained in accordance with previous research [22,23,24,25,26]. The parameters of heat-treatment are presented in Table 3.

After heat treatment. specimens were weighted at electronic weight measure device (Vesta EKS01, Nikyo Duchnice, Poland) and photographed in order to describe color and external skin structure. Then. strength of the specimens was tested. Finally, macrostructures were observed using JOEL JSM-LV SEM (JOEL Ltd., Tokyo, Japan).

### 2.2. Strength Parameters

The flexural and compressive strength parameters were measured on strength test machine (Static press C089 PN606, Matest S.p.A., Arcore, Italy) in accordance with PN-EN 1015-11:2001 standard [21]: “Methods of test for mortar for masonry. Determination of flexural and compressive strength of hardened mortar”. Test value was measured by using cold method [27]. The flexural strength analysis was conducted for three specimens from each temperature point. After that, for five half-part of the specimens compressive strength was tested. One half part of each specimen was used for microstructure research. 

### 2.3. Microstructure

Internal microstructure of prepared samples was assessed by scanning electron microscopy (SEM, JOEL Ltd., Tokyo, Japan). Specimens for observations were taken from fracture sample pieces. After sputtering with a thin layer of gold, the samples were observed using a JOEL JSM-LV SEM microscope (JOEL Ltd. Company, Tokyo, Japan) in back scattered electrons (BSE) mode at 25 kV accelerating voltage.

### 2.4. Density

Density was determined from average weight of three samples measured in different temperature points using a weight device (VESTA EKS01, Nikyo, Duchnice, Poland) and calculated specimens volume.

## 3. Results and Discussion

### 3.1. Visual Observation

Heat-treated specimens were visually assessed in order to observe changes of color and appearance of cracks and/or pores. Images of assessed specimens are presented in Table 4. The visual assessment was the first test performed after exposure at 105 °C, 150 °C, 200 °C, 300 °C, 400 °C, 600 °C and 800 °C. The main observed changes of material exposed to high temperature is color change and crack appearance. Observation shows that color did not change noticeably between 23 °C and 400 °C. The first cracks were observed above 200 °C. The patterns on the surface were like the flat figure with 5–8 sides (examples are marked in Table 4: 200 °C heat-treatment). Because of temperature increasing to 300 °C cracks propagation increased by connecting the edges of previous cracks by transverse cracks. Above 400 °C, registered changes show that thickness of existing cracks increased without increasing the amount of them. Significant changes were found at 600 °C color of specimens changed from natural beige to dark gray with black pieces. Furthermore, the number of cracks decreased. Above 800 °C, the color of the specimens changed again. from dark gray with black pieces to light gray. The size of the cracks did not increase, however, deformation of specimens was observed.

### 3.2. Density

Presented data below is motivated by determination the density of the material. The weight of samples was measured after each heating step. Figure 1 presents average density in each temperature point.

High-temperature exposure (105–800 °C) caused removal of humidity and water in pores. as well as dehydration and dehydroxylation of present cement phase of the paste in different stages of temperature. Therefore, density of the material decreased gradually to each temperature level. In the following part, density should be understood as calculated density. It needs to be emphasized that changes in value will be discussed in accordance with initial value, which is shown in Table 5.

Density was 1785 kg/m^3^ in green state (23 °C), above 105 °C density decreased by less than 1%. At 150 °C, density decreased by 3% from the green state. The first significant change in density was measured at 200 °C, where value dropped 10.7%. Above 300 °C, values sharply dropped around 22.8%. After that, density dropped slightly to 70.8% initial value (1264.6 kg/m^3^) at 400 °C. Next, at 600 °C, density fell to 1194 kg/m^3^, which was 66.9% of the starting value. Finally, last temperature point, 800 °C, shows that the value dropped to 65.5% of initial value (1169.2 kg/m^3^).

For points presented in Figure 1, two different trend line equations have been established. First, the linear trend line is dedicated for simplified calculation, where correlation R^2^ ratio was 0.85. The equation of the characteristic trend line is presented below.
y = −0.9195x + 1781.8  R^2^ = 0.8553(1)
y = ax + bb = 1781.8 ∧ ρ_23 °C_ = 1785.1 => b ≈ ρ_23 °C_(2)
ρ_T_(T) = ρ_23 °C_ + 0.92T(3)

The presented trend line equation was modified from the special equation (Equation (1)) to general equation (Equation (3)). The second order line equation (Equation (4)) is dedicated to more complicated issues, where correlation R^2^ ratio was 0.94.
y = 0.0014x^2^ − 2.062x + 1919.6  R^2^ = 0.9495(4)

### 3.3. Flexural and Compressive Strength

Flexural strength values as a temperature function are presented in Figure 2. Figure 3 shows compressive strength as a temperature function. Percentage changes of strength parameters vs initial value are presented in Table 6. 

Figure 2 shows temperature influence on flexural strength parameter of CSA paste. Firstly, flexural strength of green state of the CSA paste was f_f.CSA.23 °C_ = 2.5 MPA. Above 105 °C, flexural strength dropped to 63.3% (f_f.CSA.105 °C_ = 1.43 MPa) of initial value. Next, at 150 °C, these strength parameters f_f.CSA.150 °C_ increased to 91.5% of starting value. However, one value had significant influence on the average in this temperature point measurement, much higher than the other, where f_f.CSA.150 °C.2_ = 2.88 MPa, versus f_f.CSA.150 °C.1_ = 1.60 MPa and f_f.CSA.150 °C.3_ = 1.71 MPa. Excluding the highest value at 150 °C, the average value might be 1.65 MPa, which is still s higher value than at 105 °C. Next, at 200 °C, flexural strength decreased to 12.1% of initial value. Between 200 °C and 400 °C the values were at a similar level. At 600 °C, the value was at 6.9% of initial strength. Finally, at 800 °C, the value was f_f.CSA.800 °C_ = 0.07 MPa, which may be considered negligible.

Figure 3 presents temperature influence on compressive strength parameter of CSA paste. Compressive strength of green state CSA paste was f_c.CSA.23_
_°C_ = 61.7 MPa. Next, at 105 °C strength slightly increased to 105.3% initial value (f_c.CSA.105_
_°C_ = 65 MPa). Above 150 °C values dropped to 94.8%. Between 150 °C and 200 °C compressive strength fell heavily 37.2% (f_c.CSA.200_
_°C_ = 38.7 MPa). After that, parameter decreased to 24.4% of starting strength at 300 °C. Between 300 °C and 600 °C values were at a similar level, not less than 15% of initial value. Finally, at 800 °C, flexural strength was f_c.CSA.800_
_°C_ = 8 MPa which is 87% less than the green state of this material. Table 7 presents equations describing the trend lines as a temperature function and relation coefficient R^2^.

Based on previous results of strength tests the flexural/compressive ratio as a temperature function is presented in Table 8 and Figure 4.

Generally, F–C Ratio as a temperature function fluctuated. Initial value was 0.037. Between 23 °C and 105 °C, the ratio decreased to 0.022. After that, at 150 °C ratio rebuilt. Characteristic point, connected with fully dehydration of ettringite phase and partially dehydration of monosulfite, was at 200 °C. Owing to substantial flexural strength decrease, the F–C ratio was significantly reduced. Next, between 200 °C and 400 °C, the F–C ratio increased, because of higher compressive strength drop in comparison to flexural strength. Prior to this, between 215 °C and 270 °C aluminum hydroxide Al(OH)_3_ could be dehydroxylated, which released additional water to the structure. Between 400 °C and 800 °C, a slight decrease of the F–C ratio was observed. The trend line equation is presented with coefficient ratio R^2^.
F–C Ratio(T) = 2 × 10^−13^T^4^ − 6 × 10^−10^T^3^ − 6 × 10^−7^T^2^ − 0.0002T + 0.0417  R^2^ = 0.5781(7)

### 3.4. Microstructure 

The aim of the microstructure research was to present differences in material at different temperature points. Images of microstructures before and after heating at different temperatures are presented in Figure 5 and Figure 6.

The performance of CSA pastes exposed to high temperature is connected to their microstructure. Research indicates that micro-cracking is presented in green state material. After exposure to 105 °C, the number of defects increased. Previous research showed that at 200 °C strength parameters decreased, which might be caused by increased micro-cracking shown in the present investigation. The structure of the material became more fine and “curlier”. Pictures of structures obtained at 800 °C show a decreased amount of defects. Smaller defects probably were self-healed by a re-sintering process. Kaczmarczyk et. al [28] argue that silica with free lime and alkali particles present may create glassy phases below 900 °C. Nonetheless, some of the cracks become larger and might have contributed to the loss of load capacity. 

Map-data show that silica was located near lime (Figure 7, Figure 8 and Figure 9). These clusters of silica-lime were poor in alumina oxides content (Figure 7, Figure 8 and Figure 9). Mentioned alumina oxides were usually present near sulfur oxides. Analyses indicate that magnesium oxide created clusters with silica oxides (Figure 7 and Figure 9), where MgO may be the promoter of a re-sintering process with SiO_2_. Previous evidence shows [29,30] that magnesium oxide has a positive influence on temperature decrease of the firing process of CSA cement. The presented data and discussion indicate that during high-temperature heating (above 600 °C), the microstructure might be re-sintered. It needs to be highlighted that this creates space to modify the structure by extra materials to increase the temperature resistance of CSA paste.

## 4. Conclusions

Visual assessment might be used in general definition of the residual strength of material. CSA paste exposed to temperature lower than 600 °C did not change color significantly. The color was natural beige. Material exposed to 600 °C had a characteristic color, dark gray with black pieces. At 800 °C, the color changed to light gray. Between 600 °C and 800 °C, cracks were smaller than at 400 °C.Based on measured weight of specimens at different temperature points the equation of density as a temperature function was specified, ρ_T_(T) = ρ_23 °C_ + 0.92T with correlation coefficient R^2^ = 0.85 and ρ_T_(T) = 0.0014T^2^ − 2.2062T + 1919.6 with correlation coefficient R^2^ = 0.94. First, a linear equation might be used in a preliminary checking calculation. Second, a third order equation might be used with issues that demand more accurate calculations.The highest flexural strength collapse (12.1% of initial value) was observed at 200 °C, which relates to dehydration of main phase ettringite.The CSA paste had residual strength over entire temperature range.Compressive strength decrease was significant (24.4% of initial value) above 300 °C. Between 300 °C and 600 °C it was stable in the 15–10 MPa range (24.4–19.2% of starting value). Between 600 °C and 800 °C it was in the 10–8 MPa range (15.5–13% of green state).Microstructure and EDS investigations show that micro-cracks are present in the green state of the material. Propagation of defects was increased up to 400 °C. Above that, between 600 °C and 800 °C, number of cracks decreased. Nonetheless, defects that remained became thicker.Map-data indicate that free-lime clustered with silica. Magnesium oxide creates groups displacing other ingredients and is in close relation to SiO_2_. This might have to do with a two-component system CaO–SiO_2_, where MgO might be a sintering promotor. As a result, some micro-cracks might disappear because of re-sintering of the cement particles during high-temperature heating. Sulfonate ingredients are present in close connection with aluminium oxide.Influence of high temperature on calcium sulfoaluminate-based materials should be continued.

## Figures and Tables

**Figure 1 materials-14-06751-f001:**
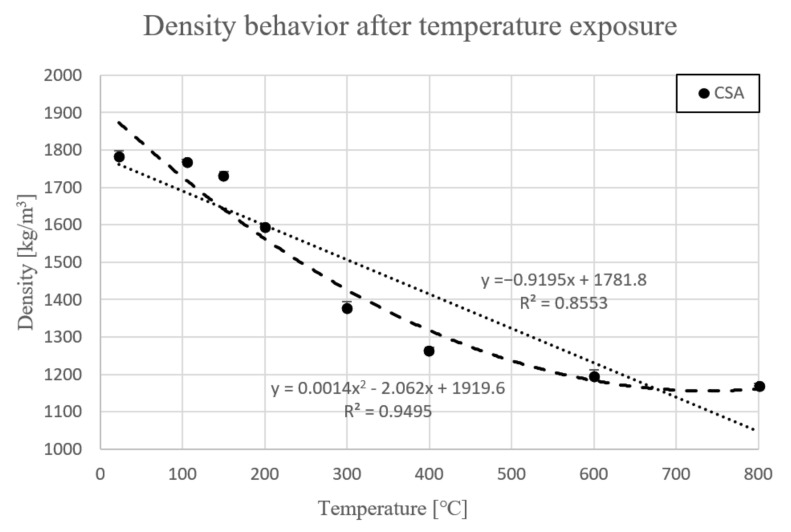
Density behavior after exposure at different temperatures.

**Figure 2 materials-14-06751-f002:**
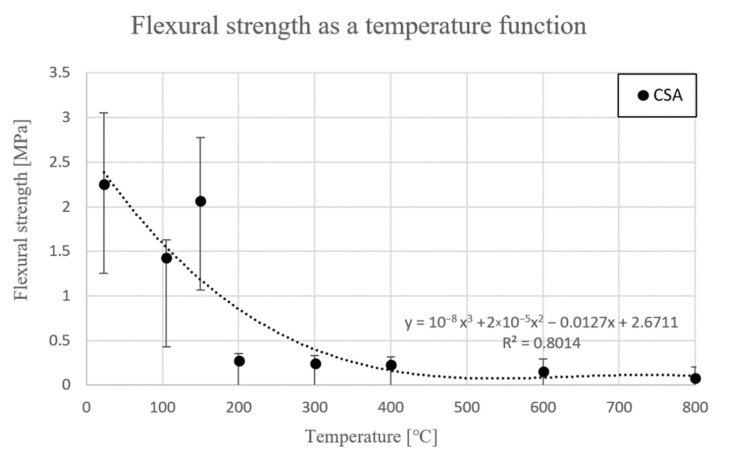
Flexural strength [MPa] in accordance with PN-EN 1015-11 as a temperature function.

**Figure 3 materials-14-06751-f003:**
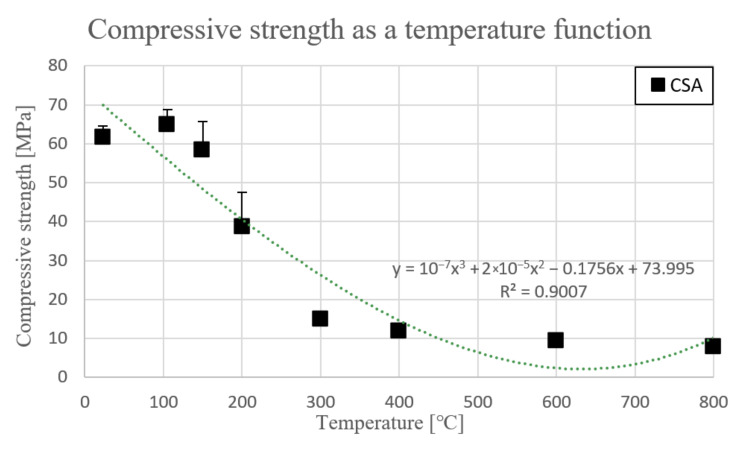
Compressive strength [MPa] in accordance with PN-EN 1015-11 as a temperature function.

**Figure 4 materials-14-06751-f004:**
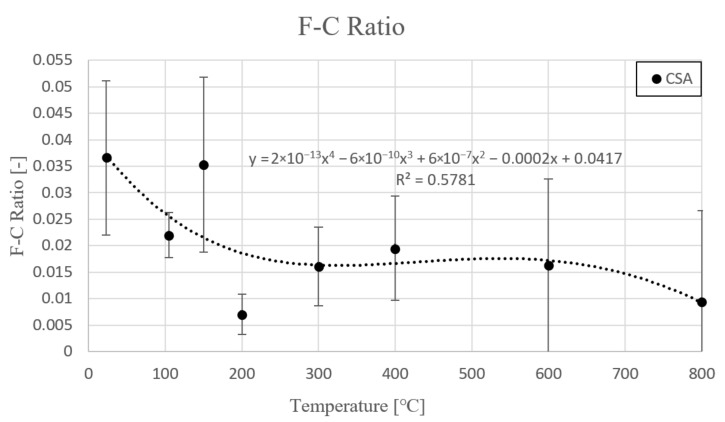
F-C ratio as a temperature function.

**Figure 5 materials-14-06751-f005:**
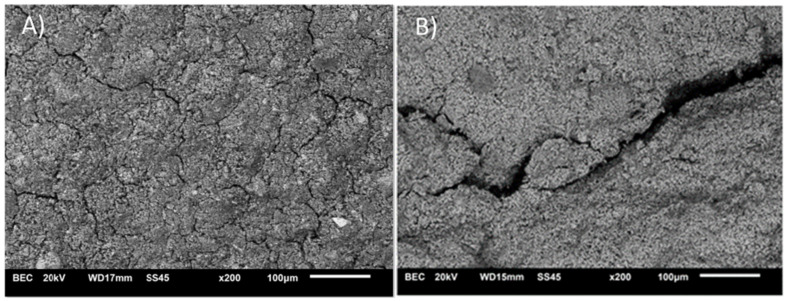
Comparison of microstructure before ((**A**) 23 °C) and after ((**B**) 800 °C) heating.

**Figure 6 materials-14-06751-f006:**
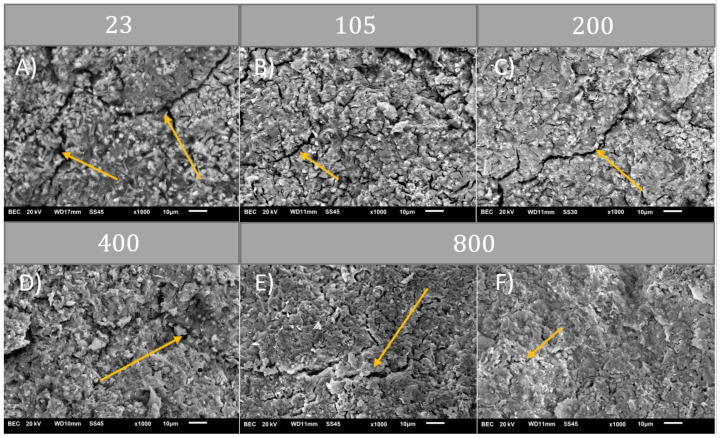
Comparission of microstructure (×1000) after heating at different temperatures ((**A**) 23 °C); ((**B**) 105 °C); ((**C**) 200 °C); ((**D**) 400 °C); ((**E**) 800 °C); ((**F**) 800 °C).

**Figure 7 materials-14-06751-f007:**
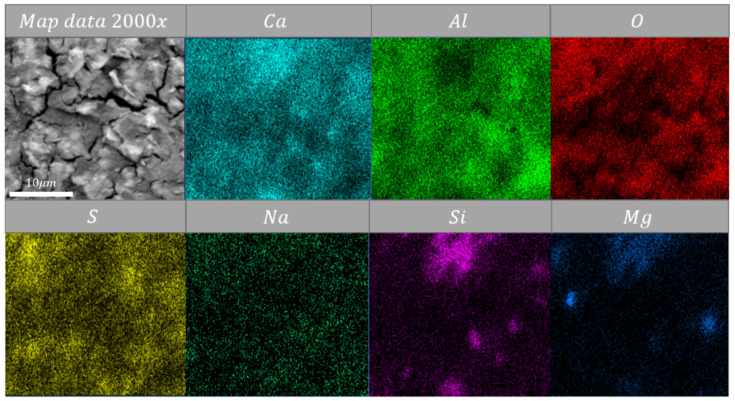
Energy-dispersive spectroscopy (EDS) map data of sample heated at 105 °C Magnification: ×2000.

**Figure 8 materials-14-06751-f008:**
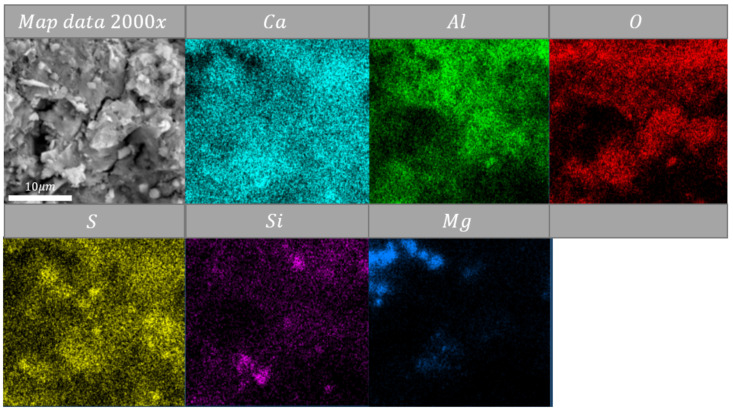
EDS map data of sample heated at 400 °C, magnification: ×2000.

**Figure 9 materials-14-06751-f009:**
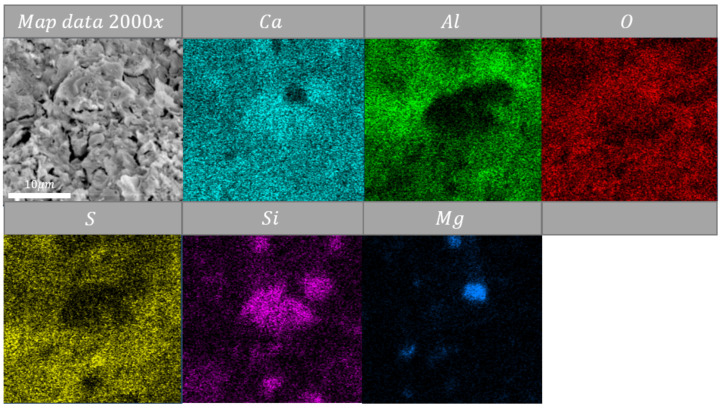
EDS map data of sample heated at 800 °C, magnification: ×2000.

**Table 1 materials-14-06751-t001:** Calcium Sulfo-Aluminate (CSA) Phase Decomposition [15,16].

Temperature Exposure	Phase Decompositions in CSA Concrete
from 90 °C	Ettringite dehydration and decomposition to monosulfite and calcium sulfate
from 150 °C	Partially monosulfite dehydration
200–300 °C	Alumina trihydrate dehydroxylation
from 450 °C	Monosulfite dehydration

**Table 2 materials-14-06751-t002:** Chemical Composition of CSA Cement.

Composition	SiO_2_	Al_2_O_3_	CaO	Fe_2_O_3_	MgO	K_2_O	Na_2_O	SO_3_	TiO_2_
wt.%	6.89	23.74	43.06	1.11	2.70	0.68	1.01	20.37	0.44

**Table 3 materials-14-06751-t003:** Parameters of Specimens’ Heat Treatment.

Heat-Treatment Temperature	Heating Rate	Isothermal Heating	Cooling Rate	Quantity of Specimens for Each Point	Specimen Dimension
105, 150, 200, 300, 400, 600, 800 °C	5 °C/min	2 h	with furnace	3	40 mm × 40 mm × 160 mm

**Table 4 materials-14-06751-t004:** Comparison of Visual Results.

Temperature [°C]		
**Raw Material**	**23**	**105**
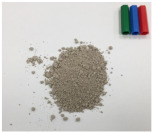	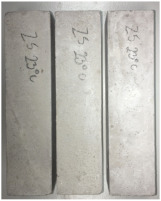	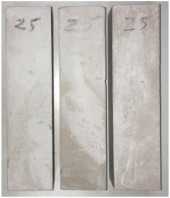
**150**	**200**	**300**
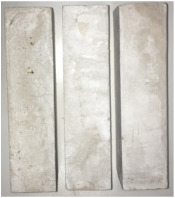	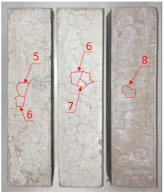	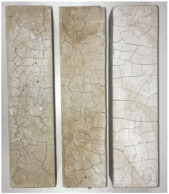
**400**	**600**	**800**
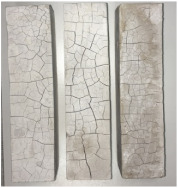	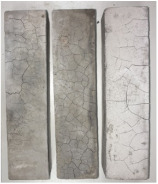	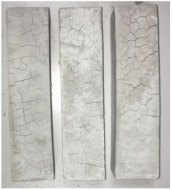

**Table 5 materials-14-06751-t005:** Change in Density of Heated Samples Relative to the Reference Sample.

Temperature °C	23	105	150	200	300	400	600	800
Density [kg/m^3^]	1785.2	1768.2	1731.8	1593.8	1377.6	1264.3	1194	1169.3
Relative density	100%	99.1%	97%	89.3%	77.2%	70.8%	66.9%	65.5%

**Table 6 materials-14-06751-t006:** Strength Parameters Percentage [%] Changes.

Temperature °C	23	105	150	200	300	400	600	800
f_f_ (%)	100	63.3	91.5	12.1	10.7	10.3	6.9	3.4
f_c_ (%)	100	105.3	94.8	62.8	24.4	19.2	15.5	13.0

f_f_—flexural strength; f_c_—compressive strength.

**Table 7 materials-14-06751-t007:** Equations of Strength Parameters’ Trend Lines.

Flexural strength	f_f_(T) = 10^−8^T^3^ + 2 × 10^−5^T^2^ − 0.0127T + 2.9711	R^2^ = 0.8014	(5)
Compressive strength	f_c_(T) = 10^−7^T^3^ + 2 × 10^−5^T^2^ − 0.1756T + 73.995	R^2^ = 0.9007	(6)

f_f_—flexural strength [MPa]; f_c_—compressive strength [MPa].

**Table 8 materials-14-06751-t008:** Value of Flexural/Compressive (F–C) Ratio as Temperature Function.

Temp. [°C]	23	105	150	200	300	400	600	800
F–C Ratio []	0.037	0.022	0.035	0.007	0.016	0.02	0.016	0.009

## Data Availability

The data that support the findings of this study are available on request from the corresponding author.

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
