# Peer review of "Impact of Elevated Temperatures on Strength Properties and Microstructure of Calcium Sulfoaluminate Paste"

_materials, 2021, doi:10.3390/ma14226751_

Round 1
Reviewer 1 Report
The manuscript presents an interesting study about the effect of high temperature exposure on the properties of CSA-based composite by means of visual assessment, microstructure, density, and mechanical strength. Overall, the present study is well written and structured. However, the following questions need to be further consideration and modification by the authors.
- The first part of Introduction needs major revision to be more focused on the main content of this study.
- Page 2, Line 79: The reasons for water treatment at 100% followed by curing at 65% humidity environment should be explained, as this is not a common curing method.
- Page 6, Line 159: “MPA” change “MPa”.
- Page 8, Line 201: The model suggested by the authors is inaccurate because it has a poorly R2 and it does not give a clear explanation on why the F-C ratio decrease firstly followed by increasing at around 200℃.
- Page 8, Line 204: It is noteworthy that microcracks can be identified in the SEM images, but unfortunately the author does not quantify these microcracks. The author should test the width of these cracks using image analysis software and try to establish the relationship between crack morphology and temperature damage.
- Page 7, Figure 4: The error bars need to be added in Fig. 4.
- Page 11, Line 251: “sTable” to be “stable”.
- Language should be further checked by a native speaker.
Author Response
Dear Reviewer,
Thank you for your comment and opinion regarding the article. As per your suggestion, we included a better explanation of the conducted research to improve the over reception of the paper.
Please see the attachment.
Yours Sincerely,
Konrad Sodol

Reviewer 2 Report
- The title should be modified into "Impact of elevated temperatures on strength properties and microstructure of calcium sulfoaluminate paste"
- Introduction is very surficial and up-to-date review is required.
- Please - improve the methodology with actual photos including raw materials, setup of tests and so on.
- Conclusion needs to be improved further.
Author Response

(The authors gave the same response as above.)

Reviewer 3 Report
The research is devoted to the investigation of the strength properties CSA-cement based composites under exposure to the heating till 800C. The main goal of the paper is to expand data base about high-temperature influence on alternative binders, like as CSA-based materials.
The cement composites specimens were prepared at base of mix CSA cement, where the main binder was mixture of CSA cement clinker, and gypsum in 4:1 ratio. Three specimens were heated 2 hours after reached 7 different temperatures 105, 150, 200, 300, 400, 600 and 800 °C.
The flexural and compressive strength parameters were defined for 3 specimens from each temperature point. Internal microstructure of heated samples was assessed by scanning electron microscopy
Comments and questions:
- Keywords repeated for calcium sulfoaluminate and CSA.
- There are a lot of points in the text, replacing commas and additional points in the paper.
- The 7 temperature points are mentioned in the paper. The 8 specimens should be tested for defining the mechanical properties. Probably it’s print mistake: “only 5 half-part of the specimens compressive strength tested”? Line 98.
- There is mistake in the Table Nr. 8 or Figure 4 for temperature 105 C.
- Could you explain more detailed the fluctuations of F-C ratio as a temperature function? Which chemical processed influence more to the compressive strength and more to the flexure strength?
- There is print mistake: sTable in 15-10 MPa range (24.4%-19.2% of starting, Line 251
Author Response

(The authors gave the same response as above.)

Round 2
Reviewer 1 Report
No further comments.